# Life-Threatening Hemorrhage from the Lingual Artery after a Genioplasty—Case Report and Review of Possible Complications Associated with Orthognathic Surgeries

**Nikoletta Vargas \*, Dasha Donado, José S. Sifuentes-Cervantes** 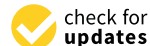**, Jaime Castro-Núñez, Lidia M. Guerrero and Luis Ferrer-Nuin**

Oral and Maxillofacial Surgery Residency Program, School of Dental Medicine, University of Puerto Rico, Medical Sciences Campus, San Juan, PR 00921, USA
* Correspondence: nikoletta.vargas@alum.urmc.rochester.edu

**Abstract:** Life-threatening hemorrhage after orthognathic surgery is rare. However, when it occurs, immediate diagnosis of the source of bleeding is imperative for successful management. The current report is the first to describe a rare life-threatening bleed from the left lingual artery following a genioplasty in a young healthy patient. Such occurrence also emphasizes the diverse anatomical course of the lingual artery and focuses on raising awareness about its numerous variations, sites of origin. Surgeons are often unaware of the anatomic variations of the lingual artery and its relation to the surrounding structures, which often lead to unexpected hemorrhages. Moreover, depending on severity, they can become challenging to manage and may lead to life-threatening complications. Taking into consideration the variations and the classification of the anatomical course of the lingual artery will improve surgical outcome, reduce morbidity, and lead to appropriate healing and recovery. This previously unpublished case regarding complications of lingual artery hemorrhage after genioplasty can bring awareness for future surgical considerations.

**Keywords:** postoperative hemorrhage; genioplasty; orthognathic surgery; lingual artery

## 1. Introduction

According to the National Institutes of Health, hemorrhage is defined as loss of blood from damaged blood vessels; it may be internal or external, and typically results in significant bleeding in short periods of time. Together, with other complications, such as postoperative pain [1], neuro-sensory deficits, inferior alveolar nerve injury, injury to other cranial nerves, infections, pseudarthroses, fractures, and amaurosis [2–6], hemorrhages after orthognathic surgical procedures can occur and can be managed successfully (Table 1). Life-threatening hemorrhages after orthognathic surgeries are rare. The incidence of intra- or post-operative hemorrhage in orthognathic surgery is higher in the maxilla, compared to the mandible [7]. For the maxilla, hemorrhages can be of venous origin arising from the retromandibular vein, or, more often, of arterial origin involving the maxillary artery or one of its branches. For mandibular osteotomies, the facial artery is the main source of hemorrhage. However, there is a paucity of reports describing a hemorrhage in the mandible from the lingual artery following genioplasty.

The lingual artery is a direct branch of the external carotid artery, which supplies the tongue, the mucosa of the floor of the mouth, and the sublingual gland. It is located deep to the hyoglossus muscle and gives off four main branches-the suprahyoid, dorsal lingual, deep lingual, and sublingual, which form an extensive anastomotic network (Figure 1). The presence of collateral vessels explains the rich blood supply of the floor of the mouth. Most postoperative hemorrhages involving the lingual artery happen after biopsies, ablative tongue surgeries, or soft tissue surgeries in general, which take place in close proximity of the aforementioned artery [8,9]. In cases of severe bleeding which cannot be managed

with mechanical pressure and packing, embolization or ligation is indicated. Ligation of its stem at the origin from the external carotid may leave a sound collateral circulation, this is the reason the lingual artery is usually ligated above the hyoid bone in order to cease severe bleeding by interruption of the entire vessel [10]. The purpose of the present article is to describe the case of a rare postoperative complication involving a life-threatening hemorrhage from the left lingual artery following a genioplasty in a healthy 29-year-old female patient. Even though three different osteotomies were performed in total, as a part of the surgical treatment plan, the article focuses on more detailed description of the genioplasty, which was performed at the end, after the Le Fort I and the bilateral sagittal split osteotomy (BSSO) were completed.

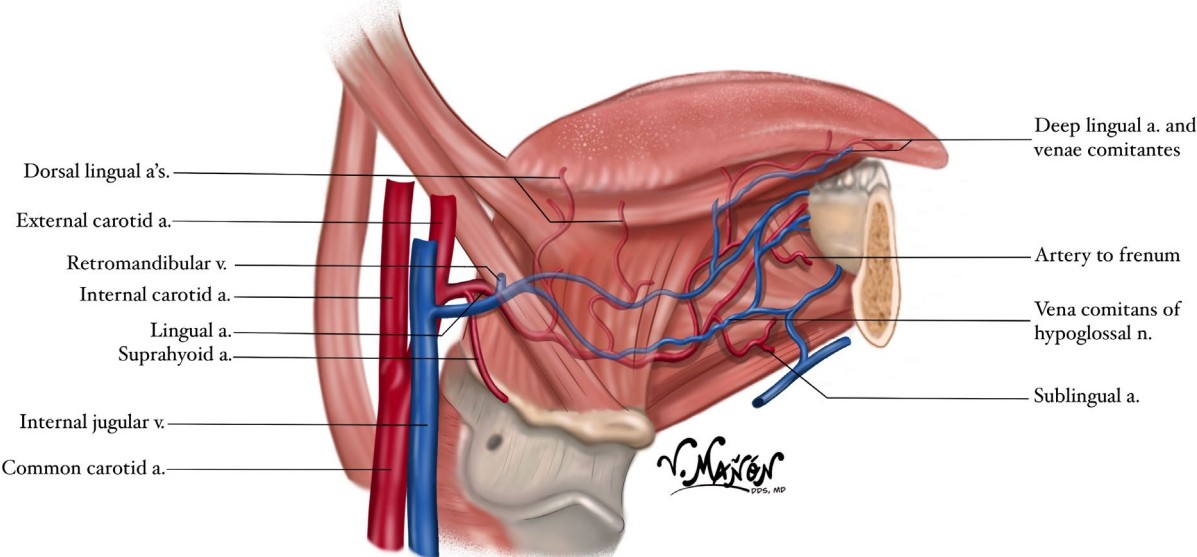

**Figure 1.** Course, branches, and surrounding structures of the Lingual Artery, Victoria Manon, DDS, MBA, MD.

## 2. Case Report

An otherwise healthy 29-year-old female patient with maxillary hyperplasia, mandibular hypoplasia and microgenia underwent Le Fort 1 impaction and bilateral sagittal split osteotomy (BSSO) advancement with genioplasty for improvement of function, phonetics, and esthetics due to a dentofacial deformity. Le Fort I and BSSO osteotomies were completed uneventfully using surgical oscillating saw. In order to correct the microgenia and improve facial balance by increasing the lower facial third, a sliding genioplasty was performed (Figure 2). All anatomic and physiologic aspects were carefully taken into consideration, and the osteotomy was performed in a standard fashion, with at least 5 mm distance from the inferior alveolar nerve and the root apices of the teeth. In order to prevent excessive notching of the inferior mandibular border, the osteotomy was extended posteriorly into the first molar [11]. During the left osteotomy of the genioplasty, brisk bleeding ensued. The bleeding was controlled with local pressure and the bony segments were rigidly fixed using a pre-bent H-shaped titanium plate. The incisions were ultimately sutured and found to be intact and hemostatic. The ceased brisk bleeding caused a soft and depressible elevation of the floor of mouth. The anesthesia team was not concerned about it, and the patient was uneventfully extubated and transferred to the post operative care unit (PACU). As the patient recovered her normotensive state, a profuse, brisk bleeding reappeared through the sutures and the left side of the osteotomy. Pressure was immediately applied to the submandibular region. Despite the bi-digital manual local pressure, the bleeding persisted. It became difficult to manage and caused an increasing non-depressible elevation of the floor of the mouth. As the tongue and floor of the mouth swelling progressed, thus jeopardizing patient's speech, and compromising her airway

(Figure 3), further packing and more aggressive pressure was applied, which ultimately led to cessation of the hemorrhage.

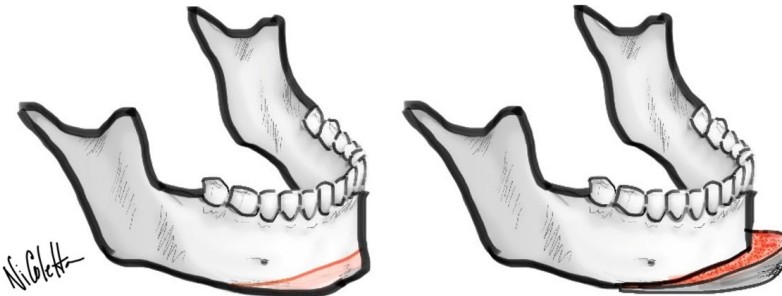

**Figure 2.** Sliding genioplasty-osteotomy pattern for correction of microgenia, Nikoletta Vargas DMD, MS, PhD.

Once the bleeding was under control, the patient was taken for a contrast-enhanced computed tomography angiography (CTA), where a left lingual artery extravasation was noted (Figure 4). Neuro-Endovascular service was then consulted, and the patient was taken for digital subtraction angiography (DSA), which was performed in a regular fashion after oro-tracheal intubation. At the time of the DSA, bleeding had already ceased (Figure 5). Due to the concerning swelling of the tongue and the floor of the mouth, a decision was made to keep the patient intubated due to possible risk of airway compromise from the sub-lingual hematoma and tongue edema. The patient's overall condition, including her vital signs and post-hemorrhagic state were closely monitored. Together with the compromised airway and the necessity to keep the patient intubated, another problematic aspect arose as a direct result from the initially uncontrolled hemorrhage due to the direct blood loss, the patient's hemoglobin decreased considerably, jeopardizing her hemodynamic stability. The total blood loss was estimated to be 1 L. As hemoglobin decreased from 12 mg/dL pre-operatively to 7.6 mg/dL after the hemorrhage occurred, the patient received a blood transfusion, which, thankfully, raised her hemoglobin to 9.9 mg/dL. The patient was successfully extubated on the second post-operative day and remained admitted for the next three days. The first, second, and fourth week follow-up demonstrated full recovery with healing as expected and no further bleeding sequelae.

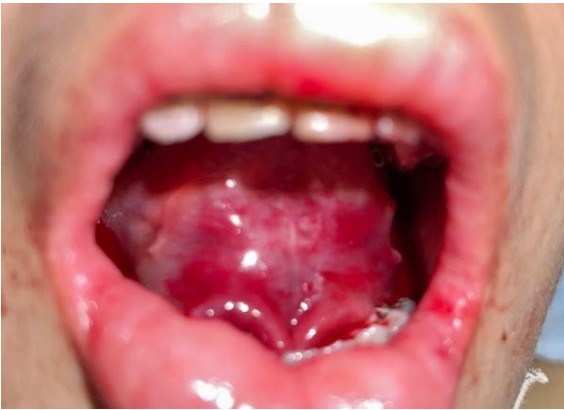

**Figure 3.** Progressing sublingual swelling prior aggressive packing and pressure.

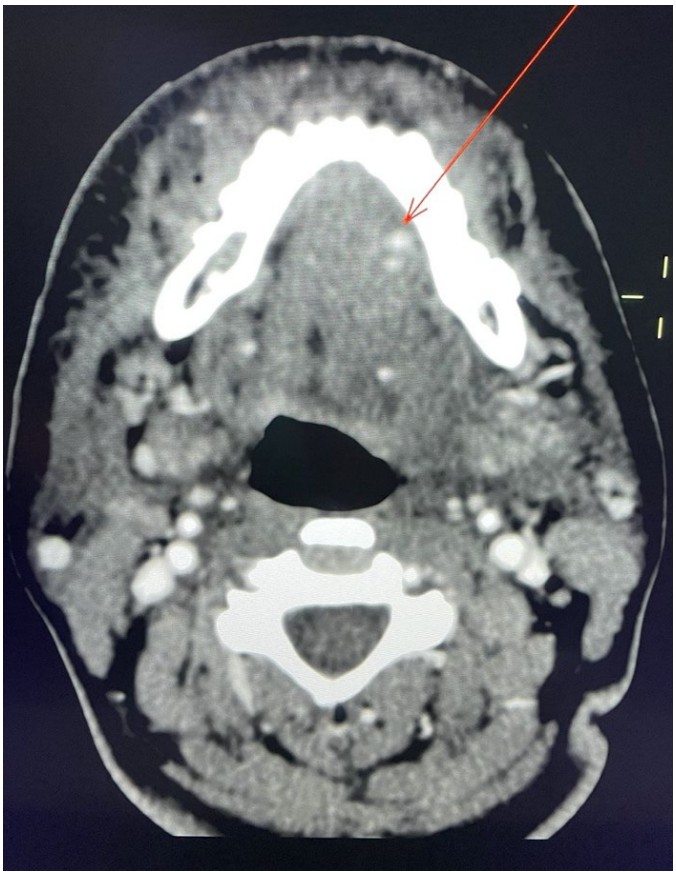

**Figure 4.** Contrast-enhanced computed tomography angiography (CTA), demonstrating a left lingual artery extravasation (arrow).

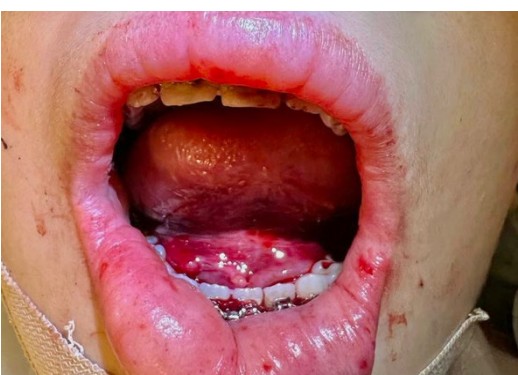

**Figure 5.** Bleeding controlled after aggressive Packing and pressure for 2 h in total.

**Table 1.** Possible complications during and after Orthognathic surgeries and other intraoral procedures. Intraoral vertical ramus osteotomy (IVRO), anterior segmental osteotomy of the mandible (ASO Md), anterior segmental osteotomy of the maxilla (ASO Mx), genioplasty (GeP), bilateral sagittal split osteotomy (BSSO), sagittal split ramus osteotomy (SSRO), bilateral sagittal ramus osteotomies (BSRO), rapid palatal expansion (RPE), bilateral vertical oblique ramus osteotomies (BVRD), fracture (fx.), and motor vehicle accident (MVA).

| Complication | Type of Surgery | Study |
|---|---|---|
| Immediate postoperative pain | IVRO, ASO Md, ASO Mx, GeP | Hsu et al. [1] |
| Inferior alveolar nerve injury | BSSO | Choi et al. [2] |
| Cranial nerve injury involving CN II, VI, ophthalmic (CN V1) and maxillary nerve (CN V2) | Le Fort I osteotomy | Kim et al. [3] |
| Upper and lower lip hypoesthesia | Le Fort I osteotomy in combination with SSRO or IVRO | Ueki et al. [4] |
| Ocular palsy due to injury of CN III and CN VI | Le Fort I osteotomy | Newlands et al. [5] |
| Mandibular infections; Mandibular pseudoarthrosis; Cranial base fractures; Hemorrhage; Amaurosis | Bimaxillary osteotomy, Le Fort I osteotomy, BSSO, GeP, Bimaxillary with GeP, Le Fort I with GeP, BSSO with GeP | Ferri et al. [6] |
| Hemorrhage: <br> Retromandibular vein <br> Aberrant venous plexus <br> Right inferior alveolar artery and vein <br> Maxillary artery <br> Maxillary artery <br> Facial artery <br> Facial artery <br> Retromandibular vein <br> Facial artery <br> Masseteric or maxillary arteries <br> Maxillary artery <br> Maxillary artery <br> Artery to mylohyoid muscle <br> Capillary, venous | Mandibularosteotomies (alone or in combination): <br> BSRO <br> Le Fort I, BSRO <br> BSRO, Reduction GeP <br> Le Fort I, BSRO, GeP <br> BSRO <br> BSRO <br> Le Fort I, BSRO, GeP <br> IVRO <br> IVRO <br> IVRO <br> Le Fort I, IVRO <br> IVRO <br> BSRO, advancement GeP <br> BSRO, advancement GeP | Lanigan et al. [7] |
| Lingual artery hemorrhage | Biopsy of a lesion in the floor of the mouth | Burke et al. [8] |

**Table 1.** *Cont*.

| Complication | Type of Surgery | Study |
|---|---|---|
| Lingual artery hemorrhage | Procedures involving carcinoma of the tongue | Li et al. [9] |
| Hemorrhage:<br>Left descending palatine artery<br>Right sphenopalatine artery<br>Descending palatine artery<br>Left pterygoid plexus<br>Left pterygoid plexus<br>A terminal branch of Left maxillary artery<br>Descending palatine or sphenopalatine artery<br>Right maxillary artery<br>A terminal branch of Left maxillary artery<br>A terminal branch of Right maxillary artery<br>Left descending palatine artery<br>Major branch of Right maxillary artery<br>Left maxillary artery or a major branch<br>Right maxillary artery<br>Maxillary artery or a major branch<br>Major branch of Right maxillary artery<br>Left sphenopalatine artery | Maxillary osteotomies (alone or in combination):<br>Le Fort I<br>Le Fort I, BSRO<br>Le Fort I<br>Le Fort I<br>Left hemi-Le Fort I with RPE<br>Le Fort I, bilateral tubinate ectomies<br>Le Fort I<br>Le Fort I, BSRO<br>Le Fort I<br>Le Fort I<br>Le Fort I, BSRO, GeP<br>Le Fort I<br>Le Fort I, BSRO<br>Le Fort I<br>Le Fort I (previous Le Fort III fx.)<br>Le Fort I, BVRO, GeP<br>Le Fort I, BVRO, GeP | Lanigan et al. [12,13] |
| Hemorrhage; Inadvertent bone-cut; Nerve damage; Inability to retroposition; Placement of condylar fragment; Non-union; Infection; Extrusion of teeth; Relapse | IVRO | Tuinzing et al. [14] |
| Varying degrees of fracture of the mandible; Displacement of the mandibular condyle out of the glenoid fossa; Bilateral neuropathy of the inferior alveolar dental nerves; Transient lingual nerve paresthesia; Slight to severe lip and face swelling with moderate respiratory embarrassment; Unusual dislocation of a proximal fragment anteriorly; Dehiscence of the wound due to hematoma formation secondary to bleeding from the osteotomy site, followed by low-grade infection and drainage; Infection; Swelling and localized osteitis in an extraction site; Subperiosteal swelling and abscess formation; Anesthetic complication with a complex fulminating pyrexia; Non-controllable postoperative nausea and vomit; Relapse of the occlusion | Intraoral sagittal osteotomy in the mandibular rami | Guernsey et al. [15] |

**Table 1.** *Cont.*

| Complication | Type of Surgery | Study |
|---|---|---|
| Near-fatal airway obstruction secondary to sublingual bleeding and hematoma possibly due to injury of submental, sublingual, and/or mylohyoid arteries | Routine dental implant placement | Niamtu et al. [16] |
| Massive epistaxis resulting in hemorrhagic shock due to a pseudoaneurysm from a distal branch of the maxillary artery, probably the sphenopalatine artery<br>Repeated epistaxis due to a pseudoaneurysm of the sphenopalatine artery<br>Epistaxis due to a small false aneurysm of the left maxillary artery<br>Chemosis of the left eye, mild left ptosis, diplopia, and an inability to move the left eye laterally from the midline due to a left VI nerve palsy secondary to extensive left carotid- cavernous fistula | Le Fort I<br><br>Le Fort I, GeP<br>Le Fort I due to a previous Le Fort II fx. in an MVA<br><br>Le Fort I with advancement and a bone graft to the alveolar cleft site due to unilateral cleft lip and palate deformity | Lanigan et al. [17] |
| Early formation of pseudoaneurysm of a lingual artery branch due to tongue trauma | Traumatic tongue bleed | Rathod et al. [18] |

### 3. Discussion

The lingual artery has been a topic of numerous studies about its variation and site of origin from the external carotid artery, common trunk with the facial artery, the superior thyroid artery, or both. Its diverse anatomical course makes it challenging for surgeons to avoid intra- or post-operative bleeding when operating in close proximity to this anatomical structure. Seki et al. [13] classified five types of lingual artery course depending on its relation to the hyoglossus and mylohyoid muscles-medial to the hyoglossus (type M), lateral to the hyoglossus (type L), transferring from lateral to medial (type T), piercing through the mylohyoid (type P), and coincident type of M and P (type C). Orthognathic surgeries, in particular genioplasties, consist of a submental and sublingual osteotomy of the anterior portion of the mandible. It is oftentimes a necessary treatment option in order to improve patient's profile by moving the chin in the desired direction. This may result in soft tissue trauma with inevitable minor blood vessels disruption. However, hemorrhage after orthognathic surgery is less common in the mandible when compared to maxilla. According to Lanigan et al. [12] it occurs in less than 1% of the cases. Most published cases describe post-operative bleeding from the facial artery, retromandibular vein, and inferior alveolar artery or vein, mainly after BSSO [14,15]. Severe bleeding can also occur from iatrogenic injuries in the floor of the mouth, or as a result of lingual perforation during implant osteotomies, as described by Niamtu et al. [16], who reported an acute respiratory obstruction and its emergency management due to a life-threatening bleeding from the sublingual artery after a routine implant placement.

The current article reports a unique case of life-threatening bleeding from the lingual artery after a genioplasty in a young healthy patient. It is unclear if this bleeding was caused by direct damage to the lingual artery during the osteotomy, or a blunt trauma due to retraction of the soft tissue during the osteotomy. The initial management for this bleeding included packing and constant mechanical pressure, for two consecutive hours. Since bleeding continued to be an issue, DSA was sought.

Intra- and post-operative bleeding can be caused due to direct mechanical trauma to blood vessels, which can be blunt or penetrating. Depending on the layer of the blood vessel that is damaged, these may respond to trauma with formation of arteriovenous fistula [17], pseudoaneurysm [18], or blood extravasation, as it occurred in this particular case. A contrast-enhanced CTA is key for locating the source of bleeding, managing and preventing severe blood loss, and avoiding further complications. The present article is the first to report a hemorrhage from the lingual artery after genioplasty. Even though a bleeding from this artery after genioplasty seems to occur rarely, the authors suggest awareness while performing this routine procedure and recommend considering this vessel as a plausible source of intra- or post-operative bleeding. More published cases and research about this surgical complication is necessary in order to better understand the occurrence and management of this life-threatening event. Orthognathic surgeons must be prepared for this kind of potential complication and have a plan of action for the treatment and successful management of hemorrhages arising from the lingual artery.

**Author Contributions:** All authors contributed to the study conception and design. Material preparation, data collection and analysis were performed by N.V., D.D. and J.S.S.-C. The first draft of the manuscript was written by N.V. and D.D., J.S.S.-C., J.C.-N., L.M.G. and L.F.-N. commented on previous versions of the manuscript. All authors have read and agreed to the published version of the manuscript.

**Funding:** This research received no external funding.

**Institutional Review Board Statement:** Not applicable.

**Informed Consent Statement:** Consent to publish has been received from all participants.

**Data Availability Statement:** Data sharing not applicable.

**Acknowledgments:** The authors thank Victoria Manon, oral and maxillofacial surgery resident at University of Texas Health Science Center for her outstanding contribution to this paper in providing the computerized color illustration.

**Conflicts of Interest:** The authors have no relevant financial or non-financial interests to disclose.

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
