# Peer review of "Life-Threatening Hemorrhage from the Lingual Artery after a Genioplasty—Case Report and Review of Possible Complications Associated with Orthognathic Surgeries"

_2673-6373, doi:10.3390/oral3010009_

Round 1

Reviewer 1 Report

* Originality / Novelty
* Significance of Content
* Quality of Presentation
* Scientific Soundness
* Interest to the readers
* Overall Merit

Author Response

Thank you for your suggestions for improvement. Please, see the attached Word file for more detailed response to your comments.

Sincerely,

Dr. Vargas

Reviewer 2 Report

1. The article states the retromandibular vein as a source of hemorrhage during le fort surgery. Are they referring to the pterygoid plexus?

2. This sentence does not make sense. If the floor of mouth is soft and depressible,  you would expect an uneventful extubation. Consider removing the word "despite"

Despite the presence of soft and 71 depressible elevation of the floor of mouth, the patient was uneventfully extubated and 72 was transferred to the post operative care unit (PACU). 

3. Does "Lingual artery" need to be capitalized?

4. The title suggests the lingual artery life threatening issue was exsanguination however it appears that potential airway embarrassment may be the "life-threating" event the authors are referring to. Consider toning down the title 

5. Consider showing post op imaging of the genioplasty to that the readers will be able to see if they cut their genioplasty in similar fashion and if they are at equal risk.

Author Response

(The authors gave the same response as above.)

Round 2

Reviewer 1 Report

Thank you for the reply and sending the modified manuscript. If you change the followings, I willingly accept the article. 

1. You have changed the title and used 'case' two times in it. Could you change again to  the other title?

2. I could not find figure legends of Fig. 3, Fig. 4 and Fig. 5. Could you indicate them?

3. You have changed  the word 'Orthognathic' to 'Facial' in line 189. I think that this will be changed to 'Oral and Maxillofacial'.

Author Response

Thank you for your comments and suggestions. Please find attached our response.

Round 3

Reviewer 1 Report

Could you add (arrow) at the end of the figure legend of Fig.4? I accept this article if this part is corrected.

Author Response

Dear Reviewer,

thank you for all your great comments and suggestions. The final correction has been made. Thank you again.

Sincerely,

Dr. Vargas
